# Preparation of Thermoplastic Cellulose Esters in [mTBNH][OAC] Ionic Liquid by Transesterification Reaction

**DOI:** 10.3390/polym15193979

**Published:** 2023-10-03

**Authors:** Elvira Tarasova, Nutan Savale, Illia Krasnou, Marina Kudrjašova, Vitalijs Rjabovs, Indrek Reile, Lauri Vares, Heikko Kallakas, Jaan Kers, Andres Krumme

**Affiliations:** 1School of Engineering, Department of Materials and Environmental Technology, Tallinn University of Technology, Ehitajate tee 5, 19086 Tallinn, Estonia; elvira.tarasova@taltech.ee (E.T.); nutan.savale@taltech.ee (N.S.); jaan.kers@taltech.ee (J.K.); andres.krumme@taltech.ee (A.K.); 2School of Science, Department of Chemistry and Biotechnology, Tallinn University of Technology, Akadeemia tee 15, 12618 Tallinn, Estonia; 3National Institute of Chemical Physics and Biophysics, Akadeemia tee 23, 12618 Tallinn, Estonia; 4Faculty of Science and Technology, Institute of Technology, Tartu University, Nooruse 1, 50090 Tartu, Estonia

**Keywords:** transesterification, cellulose esters, mTBN superbase, ionic liquids, rheology, FACE

## Abstract

The transesterification of cellulose with vinyl esters in ionic liquid media is suggested as a prospective environmentally friendly alternative to conventional esterification. In this study, various long-chain cellulose esters (laurate, myristate, palmitate, and stearate) with a degree of substitution (DS) up to 1.8 have been synthesized in novel distillable ionic liquid, [mTBNH][OAC]. This IL has high dissolving power towards cellulose, which can improve homogeneous transesterification. Additionally, [mTBNH][OAC] has durability towards recycling and can be regenerated and re-used again for the next cycles of esterification. DMSO is used as a co-solvent because of its ability to speed up mass transport due to lower solvent viscosity. The optimization of the reaction parameters, such as co-solvent content, temperature (20–80 °C), reaction time (1–5 h), and a molar ratio of reactants (1–5 eq. AGU) is reported. It was found that within studied reaction conditions, DS increases with increasing reaction time, temperature, and added vinyl esters. Structure analysis using FTIR, ^1^H, and ^13^C NMR after acylation revealed the introduction of the alkyl chains into cellulose for all studied samples. The results also suggested that the substitution order of the OH group is C7-O6 > C7-O2 > C7-O3. Unique, complex thermal and rheological investigation of the cellulose esters shows the growth of an amorphous phase upon the degree of substitution. At the same time, the homogeneous substitution of cellulose with acyl chains increases the melt viscosity of a material. Internal plasticization in cellulose esters was found to be the mechanism for the melt processing of the material. Long-chain cellulose esters show the potential to replace commonly used externally plasticized cellulose derivatives.

## 1. Introduction

Cellulose is the most important biopolymer from renewable resources to produce polymeric materials and composites. Cellulose has several beneficial properties, such as biodegradability, nontoxicity, and biocompatibility, as well as great mechanical strength, heat stability, and resistance to many solvents [1]. Some of these benefits are also drawbacks for processing: it is hard to dissolve cellulose and cellulose is not thermoplastic in its natural state [2].

Acylation of the cellulose macromolecules and substitution of hydroxyl groups of the anhydroglucose repeating units (AGU) by ester groups allows for solving the issue of material thermoplasticity. Such derivatives lose hydrogen bonding between macromolecules and obtain thermoplastic behavior [3,4]. The most important thermoplastic derivative of cellulose, widely used in plastics, films, fibers, membranes, and coatings, is cellulose acetate [5]. However, the melt viscosity of conventional thermoplastic derivatives of cellulose is still high (10^4^–10^6^ Pa·s compared to 10^2^–10^3^ Pa·s for polyolefin thermoplastics) and decomposition during melt processing may be a problem.

The cellulose esters (CEs) substituted with long chains (laurate, myristate, palmitate, and stearate) have lower melting temperatures (require less energy in processing) and better solubility in common solvents of lower polarity [6]. Such CEs exhibit plasticized polymer behavior [3]. Long-chain ester branches also have an internal plasticizing effect, which reduces or eliminates the need for added plasticizers, which are common for cellulose acetate, propionate, butyrate, or their mixed esters [7]. Thus, long-chain esters show potential as packaging and coating material with effective barrier properties.

CEs are usually prepared by esterification or transesterification procedures with different acyl donors, such as: (a) acid anhydrides; (b) acid chlorides; and (c) carboxylic acids. Long-chain fatty acids are among the most promising types of acyl donors; however, it is difficult to obtain long-chain fatty acid cellulose esters (FACEs), because of poor solubility of fatty acid compounds in polar solvent systems [8]. Strong acids as catalysts are needed as well [9].

The obstacle to acylation with acid anhydrides or acid chlorides is the formation of strongly acidic by-products, which are corrosive [10] and can reduce the degree of polymerization of cellulose. To overcome these problems, it has been recommended to use a tertiary base, such as harmful pyridine and trimethylamine [11].

In transesterification reactions, either (a) carboxylic esters [12] or (b) vinyl esters are used as acyl donors [13,14]; in this case, a large excess of the reagents and high temperature are required, and usually a low degree of substitution (DS) is obtained. The by-products of transesterification using vinyl esters as acylating agents are relatively simple compounds like alcohols, aldehydes, and ketones, which can be easily separated and recycled. Transesterification reactions have mostly been performed under heterogeneous conditions, which usually need a long reaction time (from hours to days). Furthermore, in transesterification, a catalyst is necessary to initiate the reaction [4,15], for example, sulfuric acid. In these reactions, the cellulose is not fully solubilized. Therefore, such technologies can be used mostly for the synthesis of short-chain fatty acids (with acyl chains C2–C4) [15].

Homogeneous esterification fully solubilizes cellulose and could synthesize cellulose esters of a high degree of substitution (DS) with even longer acyl chains [16]. During past decades, a large number of solvent systems were developed to dissolve and transesterify cellulose, for example, *N*-methylmorpholine *N*-oxide (NMO) [17], *N*,*N*-dimethylacetamide–lithium chloride (DMAc–LiCl) [18], dimethylsulfoxide–tetrabutylammonium fluoride (DMSO–TBAF) [19], and DMSO/KOH [15]. However, long pretreatment, long reaction times (more than 40 h), or high temperatures (>100 °C) were required, leading to relatively low DS (<2).

Those commercialized solvent systems have environmental problems; they are toxic or have low stability. Fortunately, a relatively “green” group of solvents for cellulose, ionic liquids, have been recently introduced and are under continuous development [20]. 

Ionic liquids (ILs) are polar substances and, therefore, can break the hydrogen bonds between macromolecules of cellulose. In addition, ILs are not volatile, they do not enter the atmosphere, harming humans and animals, and do not cause ozone depletion or climate change [2,21]. From this point of view, ILs can be considered green solvents, unlike DMAc–LiCl and NMO, whose broad industrial utilization is limited by instability. The number of different ILs is also nearly unlimited due to the high number of anions and cations that can be combined. Cellulose can be efficiently dissolved in many ILs and turned into a thermoplastic material by acylation in homogeneous conditions [22]. 

Processing in homogeneous conditions is one of the prospective solutions, considering commercial production of long-chain FACEs. The recent developments in IL synthesis address the durability of ILs in repeated recycling [23], combined with high dissolving power towards cellulose. Up to now, 1-ethyl-3-metyl-imidazolium acetate ([Emim][OAc]) ionic liquid was considered good for laboratory research—it has a sufficient cellulose dissolution power and can act as an in situ catalyst for the transesterification of cellulose with vinyl esters [24]. Several different CEs were prepared in [Emim][OAc] under relatively mild conditions, but an unwanted side reaction between the acetate anion of IL and the cellulose backbone has been found [25,26]. Furthermore, it is difficult to recycle the first-generation imidazolium-based ILs due to their questionable stability [27,28]. Therefore, the imidazolium-based ILs have not been used for industrial upscaling, and intensive research is going on to develop a new generation of ILs for cellulose dissolution. 

Superbases (mostly di- or triazabicyclo compounds) originating from protonic ILs are the most novel and promising solvents for cellulose dissolution and modification, for several reasons. They can dissolve cellulose to a high extent (up to 25%), have low moisture sensitivity, low toxicity, and can be recycled without degradation [29,30]. Additionally, they behave as ionic compounds with a low vapor pressure at temperatures typical for acylation reactions but can be dissociated to their superbase and acid at higher temperatures. This allows for the purification of the components by distillation [23]. 

More recently, 7-methyl-1,5,7-triazabicyclo[4.4.0]dec-5-enium acetate ([mTBDH][OAc]), and 5-methyl-1,5,7-triaza-bicyclo[4.3.0]non-6-enium acetate ([mTBNH][OAC]), were proposed to be good candidates for cellulose dissolution and regeneration [31]. The superbase mTBD demonstrates lower hydrolysis compared to DBN and the superbase mTBN shows better stability in the presence of water [23]. [mTBNH][OAC] can be recycled to either pure IL or to its starting compounds [32]. Moreover, [mTBNH][OAC] is a liquid at room temperature [33] which makes processing with this IL less labor intensive.

In this study, we focus on one of these newly developed ILs, namely [mTBNH][OAC]. We use dimethyl sulfoxide (DMSO) as a co-solvent to improve the solubility, reduce the viscosity of the reaction media, and decrease the total cost of the solvents used in synthesis. DMSO was used because it is widely available, relatively sustainable regarding production methods, recyclable, and has low toxicity [2,20].

Working with ILs as functionalization media allows precise control over DS by adjusting the reaction time, temperature, and molar ratio of reagents [12]. Control over DS also provides control over the viscoelastic properties of the produced thermoplastic materials. The degree of polymerization (DP) of cellulose, DS of the derivative, and the type of acylating agent determines the melt viscosity of the thermoplastic material and also affects elasticity, glass transition temperature [34], and thermal stability. 

CEs with aliphatic moieties C8–C18 appear thermoplastic in the DS range from 0.56 to 3. According to the literature, there is no need to produce fully substituted CEs. A DS ≥ 1.5 is high enough since CEs with a DS between 1.5 and 3 show similar mechanical and thermoplastic properties [35,36]. 

The plasticizing effect of aliphatic chains was found to be dependent more on the DS than on the fatty chain length, but no clear influence of the DS on thermal transitions was observed [36]. No glass transition T_g_ in cellulose palmitate is reported in the paper [37]. Jinlei Li in his work discovered a correlation between T_g_ increase and the formation of materials with solid-like behavior in a molten state. At the same time, in contradiction to previous research, it is stated that T_g_ is influenced more by acyl side chain length than the DS [38]. Additionally, CEs with short alkyl chains were found to exhibit a narrow thermoplastic window between melt flow and thermal degradation [39]. Recent developments show that long-chain FACEs are promising but underdeveloped thermoplastic materials with complex relationships between properties and structure. 

In this work, we investigate the optimization of the homogenous transesterification of cellulose with long-chain fatty vinyl esters (VEs) in distillable IL, [mTBNH][OAC], without any additional catalysts and under mild conditions, to produce thermoplastic FACEs by a sustainable and green procedure. This will complete the picture of [mTBNH][OAC] application for cellulose modification, as far as it has mostly been studied for cellulose regeneration [29], and for preparation of cellulose acetate [40].

Another goal of this study is to disclose the melt flow behavior and the rheology of long-chain FACEs, which are not widely presented in the literature. Melt properties and behavior are important for understanding the proper processing conditions for blow molding, injection molding, sheet forming, extrusion, fiber spinning, etc. Herein, we further develop the main concepts of our previous rheological studies discussed [41,42]. 

## 2. Experimental Section

### 2.1. Materials

Microcrystalline cellulose (MCC) by Carl Roth GMBH (Karlsruhe, Germany). Vinyl laurate, vinyl myristate, vinyl palmitate, and vinyl stearate with purity > 98% were purchased from Tokyo Chemical Industry Co. (Tokyo, Japan). Ionic liquid 5-Methyl-1,5,7-triaza-bicyclo-[4.3.0]non-6-enium acetate, [mTBNH][OAC], was not commercially available and was synthesized by Liuotin Group Oy (Porvoo, Finland). The melting point of IL is 15 °C; the flash point is more than 220 °C. DMSO with a purity of 99.9% was purchased from Fisher Chemical (Pittsburgh, PA, USA). Low-density polyethylene (LDPE), Petrothene NA960000 supplied by LyondellBasell (Rotterdam, Netherlands). Diethyl phthalate plasticized (24% DEP) cellulose acetate (Mazzucchelli PCA) supplied via S. e P. Mazzucchelli (Castiglione Olona, Italy). Pyridine-d_5_ (purity = 99.5 atom % D. contains 0.03 *v*/*v*% TMS) and Chloroform-d (purity = 99.8 atom % D. contains 1 *v*/*v*% TMS) for NMR were purchased from Acros Organics (Geel, Belgium).

### 2.2. Cellulose Dissolution and Esterification Procedure

MCC was dried under a vacuum at 105 °C for 24 h before use. Cellulose concentration in solutions was 2–5%. 2–5 g of MCC (12.3–30.7 mmol for the AGU) was dissolved in 100 g [mTBNH][OAC] (505 mmol) and stirred at 60 °C for 24 h until the cellulose was completely dissolved to yield 2–5 wt%-solution. To decrease the viscosity, a co-solvent DMSO was added. In the case of the use of co-solvent, IL and DMSO have been first mixed and then cellulose has been added to this binary solvent. Different ratios of DMSO/IL have been prepared and studied. The designed amount of the respective vinyl ester (from 1 up to 6 equivalents to cellulose anhydroglucose unit (eq./AGU)) without any external catalyst was added to the cellulose solution in a chemical reactor equipped with a mechanical stirrer and nitrogen flow, and then the reaction was performed at designed conditions under nitrogen atmosphere and intensive stirring. When the reaction was completed, the obtained cellulose esters were precipitated into 500 mL of water (for laurate and myristate) or ethanol (for palmitate and stearate). To remove solvent and vinyl ester residuals, the product was washed several times in 100–200 mL of ethanol, then acetone, and with hexane. Finally, the product was dried under a vacuum at 55 °C overnight.

For the prepared products, transesterification of CEs was repeated 2–3 times to verify the reproducibility of the reaction. It was found that DS deviates within 5–10% for the batches.

### 2.3. Characterization

The FT-IR spectra of MCC and cellulose derivatives were recorded on Interspec 200-X using the Quest ATR accessory purchased from Specac Ltd (Orpington, UK) in the range of 500–4000 cm^−1^ with a resolution of 4 cm^−1^. All samples were dried at 80 °C in a vacuum oven overnight before the FTIR measurement to remove the moisture. 

The ^1^H NMR, ^13^C NMR, COSY, and HSQC NMR spectra of cellulose derivatives were acquired on an Agilent Technologies DD2 500 MHz spectrometer equipped with 5 mm broadband inverse (^1^H, HH-COSY, HC-HSQC spectra) or broadband observe (^13^C spectra) probes. A 15 min temperature equilibration delay was allowed between sample insertion and NMR acquisition at 40 °C (cellulose palmitate and stearate in CDCl_3_) or 80 °C (cellulose laurate in DMSO-d_6_ and cellulose myristate and laurate in Py-d_5_) sample temperature. Typically, for ^1^H spectra, 32 scans with 25 s relaxation delay were acquired and for ^13^C 20,000–45,000 scans with 2.5 s recycle delay were acquired to achieve the desired signal-to-noise ratio. An HMBC spectrum for sample CP-3 was acquired in 16 h on an 800 MHz Bruker Avance III spectrometer equipped with a He-cooled cryoprobe.

The NMR samples were prepared in a small glass bottle where 20–25 mg of the sample was dissolved in 0.8 mL of deuterated NMR solvent for 90 min at 40–65 °C until a clear solution was obtained. The glass bottle containing the mixture was dried and carefully sealed with parafilm. The samples were then subjected to ultrasonic treatment to obtain a transparent solution. 

The DS of cellulose laurate, -myristate, -palmitate, and -stearate was calculated from the ^1^H NMR spectrum by taking an integral of the area of terminal methyl groups (*I_(_*_CH3)_) and AGU signals (*I*_AGU_) based on the reported method [43]:(1)$$DS=10\cdot \frac{I_{\left({CH}_{3}\right)}}{\left(3\cdot I_{AGU}+I_{\left({CH}_{3}\right)}\right)}$$

Additionally, the DS of obtained samples was also verified by the standard saponification ASTM D871-96 method. The DS was calculated according to [34]:(2)$$DS=\frac{162\;E}{100M-E\cdot (M-1)}$$
where *E* is the ester content obtained by the saponification method, and *M* is the molecular weight of the grafted acyl residue.

To study the rheological properties of samples, an Anton Paar Physica MCR501 rheometer, with cone-plate measuring geometry was used. To prevent oxidation, the experiments were carried out in a nitrogen atmosphere and did not last longer than 10 min at a temperature of 190 °C. Flow curves were obtained at a shear rate $\dot{\gamma }$ range from 0.01 to 100 s^−1^. Complex viscosity was determined in the range of angular frequencies ω = 0.01 to 500 rad/s. A constant strain of γ = 5% was determined from the linear viscoelastic region (LVR). An amplitude sweep test to determine the LVR was conducted at a frequency of 1 Hz. Samples for rheology were prepared from 100 µm thick films cast from solution in chloroform or pyridine at room temperature overnight by cutting 2.5 mm Ø discs. Non-soluble materials were compressed by a hydraulic press into 100 µm thick and 2.5 mm Ø discs in a stainless-steel mold at a pressure of 70 bar at room temperature.

A differential scanning calorimeter (Perkin Elmer DSC-7) was used for the thermal analysis of materials at scanning rates of 30 °C/min. Nitrogen was used to purge the furnace. To avoid differences in melting and crystallization temperatures caused by variations in sample weight, a sample mass of 5.00 ± 0.02 mg was used in all DSC experiments. Film samples were packed into aluminum foil to maximize thermal contact between the sample and the calorimetric furnace. DSC samples were prepared from solvent-cast 100 µm thick films. Non-soluble materials were compressed by the hydraulic press into 100 µm thick flakes in a stainless-steel mold at a pressure of 70 bars at room temperature.

The size exclusion chromatography (SEC) curves of cellulose esters were determined using gel permeation chromatography (GPC) on the Shimadzu Prominence system equipped with a Shodex KF-804 column and a refractive index detector (RID-20A). Calibration of the GPC system was performed using 3 separate polystyrene standards (74,800 Da, 230,900 Da, and 473,600 Da). CE samples (10 mg) were dissolved in pyridine (1–2 mL) and the GPC analysis was conducted at 60 °C using pyridine as the mobile phase with a flow rate of 0.5 mL/min. Molecular weights (number-average *M*_n_ and weight-average *M*_w_, polydispersity index PDI = *M*_w_/*M*_n_) of cellulose esters have been calculated. Several samples were tested 2–3 times and the obtained standard deviation of MM measurements was 3–8%.

Molar mass (MM) of pure MCC was determined at 25 °C from the intrinsic viscosity [η] of cellulose solution in cupriethylenediamine hydroxide solution, Cuene, according to a standard procedure [44]. The MM was then calculated by the Mark–Houwink equation with parameters K = 1.01 × 10^−4^ dL/g and a = 0.9 [45]. The obtained MM was 163,000 g/mol.

## 3. Results and Discussion

### 3.1. Preparation of Cellulose Esters

Figure 1 shows the pathway for cellulose transesterification with vinyl esters. Cellulose esters were synthesized in one step and low-vapor-point ethanol (in situ tautomerization to acetaldehyde) as the byproduct can be easily evaporated from the reaction system at elevated temperatures.

It is well known that ILs are efficient solvents for the dissolution and modification of cellulose; however, there are also some drawbacks, such as high cost and high viscosity. For example, in the current work, it was possible to dissolve cellulose in pure IL only at 2 wt% concentration—higher concentrated solutions were not possible to stir effectively due to high viscosity. To scale up the entire production process, the cellulose content should be raised. This drawback can be overcome by using the mixtures of IL with cheaper and less viscous co-solvents. A dipolar aprotic solvent such as DMSO is an effective co-solvent for cellulose dissolution. The addition of DMSO to IL can enhance its solvating ability because DMSO can speed up mass transport due to lower solvent viscosity. Indeed, if the content of DMSO is increased, the viscosity of cellulose solution in binary IL:DMSO solvent drops significantly (see Table 1 and Appendix A) at all studied temperatures (including temperatures at which the transesterification reactions were conducted) while the concentration of cellulose in all tested solutions was fixed at 2 wt%. As is seen in Table 1, the viscosity of cellulose solution in 1:3 IL:DMSO binary solvent is approximately seven times lower than in pure IL. Surprisingly, even 25% of IL in binary solvent (i.e., 1:3 IL:DMSO ratio) is sufficient to dissolve cellulose, and optically transparent solutions of cellulose are observed. This indirectly confirms the high efficiency of novel IL to dissolve cellulose. Taking into account such low viscosity of cellulose in a binary solvent, it is possible to increase the cellulose solution concentration from 2 wt% up to 5 wt% and even more, keeping the same mixing level. Hence, the consumption of IL can be decreased while the level of CE production can be increased 2–3 times.

Several reactions have been conducted at different IL:DMSO ratios, while the reaction conditions were kept constant, (T = 60 °C, t = 2 h, 3 eq./AGU). Vinyl laurate (VL) was chosen as an acylation agent. The results of this study are presented in Table 1. DMSO is not a solvent for cellulose; an increase in DMSO content could limit the ability of cellulose toward homogeneous transesterification. This could explain the DS and viscosity drop for CL-4 and CL-5 when the IL:DMSO ratio exceeds 1:1.

Structure analysis utilizing FTIR after acylation revealed the introduction of the alkyl chains into cellulose for all studied samples. FTIR spectra of native cellulose and its laurate derivatives were obtained in solvents with different IL:DMSO ratios (see Appendix A). The strongest evidence of successful acylation is the appearance of an absorption band at 1746 cm^–1^ that corresponds to the stretching of the ester carbonyl (>C=O) group. Compared to unmodified MCC, there were several new peaks in the FTIR spectra of modified samples. New peaks at 2950 cm^−1^ and 2847 cm^−1^ correspond, respectively, to asymmetric and symmetric stretching of the methylene group present in cellulose laurate (CL). The presence of these peaks indicates that the methylene and carbonyl groups were attached to cellulose, and hence, successful transesterification of cellulose in the novel [mTBNH][OAC] IL mixed with DMSO, without external catalyst, has been conducted.

All produced CLs are soluble in several organic solvents, mostly DMSO, THF, and pyridine, which enables direct measurements of NMR spectra in the form of solution and, thus, provides a tool for the calculation of DS. The chemical structure of the cellulose laurate (CL) was confirmed by ^1^H NMR and ^13^C NMR. In the ^1^H NMR spectrum of the CLs (See Appendix A), the proton signals from approximately 6.00 to 3.50 ppm are assigned to H-1, H-2, H-3, H-4, H-5, H-6, and H-6′ of AGU in cellulose. The signals at 2.32–1.93, 1.88–1.45, and 1.27 ppm are associated with the methylene protons at H-8, H-9, and H-10–17, respectively. The signals at 0.98–0.79 ppm are attributed to the methyl protons at H-18.

In the ^13^C NMR spectrum of cellulose laurate, presented in Figure 1, the signals at 34.78, 32.61, 25.80, 23.34, and 14.60 ppm are assigned to carbons of C-8, C-16, C-9, C-17 and C-18 of the aliphatic side chain, respectively. The carbons at C(10-15) give signals starting from 30.44 to 30.04 ppm. The AGU carbons C-1, C-1′, C-4, C-2,3,5, C-6, and C-6′ give signals at 104.98, 102.33, 81.50, 76.91–74.35, 64.32 and 62.68 ppm, respectively. The signals from 173.89 to 170.98 ppm correspond to the carbonyl carbon at C-7, which provides the direct confirmation of the successful attachment of the long-chain fatty acid chain to the cellulose backbone. Three peaks for the carbonyl group indicate that all OH groups at positions 2, 3, and 6 were acylated [19]. According to the ref. [46], by peak integration of carbonyl carbons (C7-O2, C7-O3, and C7-O6), it can be suggested that the partial DSs of C7-O6, C7-O2, and C7-O3 are 0.41, 0.11, and 0.08, and, consequently, the substitution order of the OH group is C7-O6 > C7-O2 > C7-O3. 

The data of DS obtained from NMR spectra are presented in Table 1. Although the cellulose solution viscosity decreases with increasing DMSO content in a binary solvent, the DSs of samples are not much changed until the ratio 1:1 IL:DMSO. The further increase in DMSO content decreases the DS of CLs 1.5 times. Therefore, it can be concluded that DMSO content exceeding the amount of IL in a mixture, causes obstacles in the transesterification process of cellulose with vinyl esters. Therefore, for further experiments, the optimum 1:1 IL:DMSO ratio was chosen for cellulose dissolution and modification. This helps to decrease the cost of production, due to the lower IL content used and the higher yield of the cellulose derivatives.

As is seen in Table 1, CLs synthesized in the above-mentioned reaction conditions have rather a low DS. However, it should be noted that in our study DS = 0.6 was obtained during 2 h of reaction at a moderate temperature of 60 °C, whereas similar DS values have been reported by Wu et al. [47] but the reaction time was 24 h. No external catalyst or activation was required to achieve this reactivity in [mTBNH][OAC]. Partial decomposition of such ILs during the acylation process liberates the superbase [48], which can act as a basic catalyst to promote the transesterification reaction analogically to 1,8-diazabicyclo[5.4.0]undec-7-ene (DBU) [47], 1,5,7-triazabicyclododecene (TBD) and 1,5-diazabicyclo(4.3.0)non-5-ene DBN [49]. TBD and DBU are also known as superbases, which give protonic ILs in combination with an acid component. The organocatalytic effect of DBU and TBD during transesterification was demonstrated by several authors [47,49,50].

Several attempts to improve the degree of substitution have been made. The results are presented in Table 2. 

Additionally, SEC measurements of the different CLs were performed, as the cellulose esters were all soluble in pyridine, except CL-6. SEC curves of studied CLs are presented in Appendix A. 

To investigate the effect of the reaction time on the transesterification, the reactions were carried out at 2 (CL-3), 4 (CL-8), and 5 (CL-9) hours at the same temperature of 60 °C and AGU:VL = 1:3. Analysis of the reaction products reveals that the DS can be increased by employing longer times of reaction. For example, the DS obtained at 4 h of reaction (CL-8) is approximately 2.6 times higher than the DS obtained at 2 h of reaction (CL-3, see Table 2). At 5 h of reaction, the DS is even higher, achieving values of 2.0 (CL-9). However, starting at 4 h the color of the reaction medium started changing from dark yellow (at 4 h) and even brown (at 5 h). The color change can be a sign of either IL or cellulose degradation. To clarify, IL has been held for 3 h at different temperatures: room temperature, 60 °C, 70 °C, 80 °C, 100 °C, and even 110 °C. No signs of IL degradation have been detected. FTIR spectra of all ILs treated at different temperatures were identical; color has not been changed. It can be concluded that IL is stable at the studied temperatures.

However, SEC data (see Table 2) of cellulose laurates show that samples undergo some degradation with prolonged reaction times. Taking into account the viscosity-average MM of cellulose (163 kDa), MM of vinyl laurate (226 Da), and definite degree of substitution, it is possible to estimate the molar mass of cellulose laurate at certain DSs (note that viscosity-average MM is always a little bit higher than M_n_). The estimated molar mass of CL-3 is close to the experimentally obtained one. However, the estimated MM of CL-8 should be around 500 kDa, while the *M*_n_ of CL-8 is only 358 kDa. Experimentally obtained *M*_n_ of CL-9 decreases down to 304 kDa, which is twice lower than the estimated MM (around 600 kDa) and 1.2 times lower than CL-8.

The effect of reaction temperature on DS was investigated by using three temperatures for the transesterification, while the time of reaction (2 h) and molar ratio AGU:VL (1:3) were kept the same. In the first reaction, 20 °C was used for the transesterification, yielding degrees of substitution equal to zero—no transesterification at room temperature was possible. Further reactions were carried out at 60 °C, revealing that an elevation of the temperature leads to successful transesterification of cellulose with a DS = 0.6. The increase in the DS can be explained by the higher flexibility of the cellulose chain, and easier removal of acetaldehyde as a by-product of the reaction. The last reaction was conducted at 80 °C, giving cellulose laurate with a DS = 1.2. A similar effect of temperature was detected for cellulose palmitates. However, at 80 °C and higher temperatures, the reactive solutions turned dark brown, indicating the hydrolysis process of IL and/or cellulose esters. SEC data (see Table 2) showed that compared to the unmodified MCC, the molecular weights *M*_n_ of the acylated samples increased with increasing DS. However, samples undergo degradation, both with prolonged reaction times and at high reaction temperatures ≥80 °C. The MM of CL-7 obtained at 80 °C is smaller than that of CL-3 synthesized at 60 °C. The same is true for samples prepared at 3 h of reaction and different temperatures; MM of CL-11 is approximately 1.7 times smaller than that of CL-10 (see Table 2). Right away, we have no suitable explanation for this chain degradation. Somewhat analogically to [DBNH][OAc] ionic liquid [49], it can be suggested that degradation of the cellulose backbone starts with the generation of acetic acid, which initially was entrapped by the excess of mTBN.

Therefore, for the transesterification process in [mTBNH][OAC] temperatures of reaction lower than 80 °C should be employed. The optimum conditions were chosen as 70 °C and 3 h, which give the highest DS without degradation. However, it should be stressed here that in our case the degradation of cellulose esters is not a big issue. Taking into account very high molar masses of final products, some decrease in MM of cellulose esters can even be an advantage for processing and thermoplastic properties of products.

The chosen optimum reaction conditions were applied to the synthesis of longer (than laurate) FACEs, namely cellulose myristate (CM), cellulose palmitate (CP), and cellulose stearate (CS). The parameters of synthesis are listed in Table 3. Structure analysis of all samples using NMR and FTIR after acylation discovered the attachment of the alkyl chains onto cellulose. ^1^H /^13^C NMR spectra of CM, CP, and CS samples are presented in Appendix A.

To confirm further the assignment of the signals of cellulose esters, the HSQC spectrum was collected. The example of ^1^H-^13^C HSQC for cellulose palmitate is presented in Figure 2, where the cellulose region (Figure 2a) and aliphatic side chain region (Figure 2b) are shown. As can be seen, in the aliphatic side chain region, the correlation of C-21/H-21, C-20/H-20, C-22/H-22, and C-10–19/H-10–19 were located at δH/δC at 1.29/22.54, 1.27/31.76, 0.89/13.90 and 1.26/29.46, respectively. In the cellulose region, the strong correlations were distinguished at δH/δC 5.14/72.17, 4.82/71.93, 4.42/101.59, 4.19/62.34, 3.52/73.06, and 3.36/82.31 for C-3/H-3, C-2/H-2, C-1/H-1, C-6/H-6, C-5/H-5, and C-4/H-4, respectively. The two decentralized signals of C-8/H-8 and C-9/H-9 are represented by δH/δC at 2.32/33.80 and 1.59/24.78, respectively.

Figure 2 reveals that under optimized reaction conditions, complete acylation of C6-O is achieved as only a single resonance of the corresponding group in ^13^C. Additionally, the intensity of the characteristic cross-peak of C2-OAcyl (4.8–4.9/72 ppm) is greater than that of C3-OAcyl (5.1/72 ppm), thus indicating a preferential acylation.

An estimation of acylation efficiency at different positions in the CP was performed by comparing respective signal integrals from HSQC spectra. Integration of the corresponding signals in the HSQC spectra showed that a total degree of substitution (DS = 1.80, see Table 3) has, within a reasonable error margin, ratios of 1.00: 0.47: 0.33 for C7-O6: C7-O2: C7-O3 positions, respectively. Consequently, similarly to CL, the substitution order of the OH group in CP is C7-O6 > C7-O2 > C7-O3.

The ^1^H-^1^H COSY spectra, presented in Figure 3, revealed the correlation of proton-proton signals. Figure 3b indicates the coupling between hydrogens of the aliphatic side chain region. Cross peaks at frequencies 1.26/0.87 and 0.87/1.29 show coupling between CH_3_/22 and CH_2_/10–19. Cross peaks at frequencies 1.57/1.32 and 1.27/1.60 show coupling between the peaks CH_2_/10–19 and CH_2_/9. Similarly, cross peaks at frequencies 2.31/1.60 and 1.62/2.31 show the coupling between CH_2_/9 and CH_2_/8. 

In the AGU region (Figure 3a), the spectrum is more crowded, as many correlations for various combinations of non/acylated components are present. Thus, two correlations of the glycosidic H-1 and acylated H-2 are visible at 4.4/4.8 and 4.4/4.9 ppm, depending on the substituent at O-3. The correlations of the H-2 with acylated and unsubstituted H-3 are observed at 4.9/5.1 and 4.8/3.6 ppm, respectively. Correlations of H-3 and H-4 are at 5.1/3.8 and 3.6/3.7 ppm, those of H-4 and H-5 are observed at ~3.7/3.4, and those of H-5 and acylated H6 and H6′ are at 3.4/4.3 and 3.4/4.0 ppm, respectively.

Heteronuclear multibond correlation spectroscopy (HMBC) was also conducted to analyze the hydroxy (-OH) substitution at the cellulose backbone. The HMBC spectrum (see Appendix A) showed the correlations between the acyl carbon (C-7) signal at 170.95 ppm and H-3 proton signal at 4.84 ppm and H-2 proton signal at 4.77 ppm, additionally proving the esterification of positions C7-O3 and C7-O2, respectively. Unfortunately, the correlation between C-7 and H-6 protons of AGU was not detected. However, according to the literature, this correlation peak is much weaker and usually not detected by HMBC for cellulose esters [51].

As can be seen in Table 3, transesterification of cellulose in [mTBNH][OAC] with longer vinyl esters was successfully conducted and a DS up to 1.8 has been achieved. A similar DS was obtained by T. Kakko et al. [49], but for cellulose acetate and propionate in another distillable IL, namely [DBNH][OAc]. For long-chain CEs, it was possible to obtain similar values of DS but in DMAc/LiCl [52], which is an undesirable solvent for a future greener life.

By comparing the DS for the different cellulose esters, it can be said that under the same reaction conditions, the DS and, as a consequence, *M*_n_ of the cellulose esters decrease with increasing fatty acid chain length. Such a result is expected due to the higher steric hindrance of long ester chains and it has been demonstrated in the literature [12]. Cellulose laurate, myristate, and palmitate show DS > 1, which makes them soluble in many organic solvents. However, cellulose stearate CS-1 shows approximately twice the lower values of DS and as a result, CS-1 is insoluble in any tested solvents. 

DS is the most important property determining the thermoplastic behavior of CEs, and, as it was explained before, the DS value should be close to or above 1.5 to expect similar processing behavior as commodity polymers have. For cellulose laurate and myristate, it was possible to obtain a DS ≥ 1.3; however, for cellulose palmitate and stearate the reaction conditions should be modified to achieve a higher DS. DS can further be increased by increasing the temperature of the reaction. Indeed, for cellulose palmitate CP-3 prepared at 80 °C, the DS has been improved up to 1.8. However, instead of an expected increase in *M*_n_, CP-3 demonstrates a drop in molecular weight, which can be attributed to the cleavage of the cellulose backbone.

Degradation of cellulose backbone during transesterification reaction at temperatures ≥80 °C can be revealed also by measuring intrinsic viscosity [η]. As it is well known, the intrinsic viscosity of the polymer directly depends on its molar mass; the higher the [η] the higher the MM. The intrinsic viscosity of CP-1 in pyridine at 20 °C was 2.5 dL/g, CP-2 has an intrinsic viscosity of 2.8 dL/g, whereas CP-3 has [η] = 1.1 dL/g, which is more than two times lower and confirms the cellulose degradation.

It is well known by Le Chatelier’s Principle that an excess of one reactant will drive the reaction to the right, increasing the production of ester, and finally increasing the yield of ester. Therefore, in subsequent steps for preparation of CP and CS of higher DS without degradation, the reactions were carried out using excess vinyl esters, namely molar ratios AGU:VE = 1:5 and 1:6 were employed. As can be seen in Table 3, the degree of substitution of CS (CS-2) was improved more than two times, and DS = 1.4 was achieved when 6 eq/AGU were taken. For CP the DS = 1.30 was already obtained at 5 eq/AGU. Further, an increase in the ratio of the vinyl esters probably will allow for achieving a higher DS, but an unreacted vinyl ester and its by-product stay in a supernatant phase and make purification of end-product and IL recycling difficult. 

Generally, the cellulose esters with DS varying from 0.6 (and probably lower) up to 2.0, depending on the reaction parameters and the vinyl ester used, can be produced. It should be noted that the reaction conditions used in the current work are much milder than those which are used usually in transesterification reactions in other ILs, where higher temperature, longer reaction time, and the presence of a catalyst are required to achieve an appropriate DS [4,15]. The developed method demonstrates the first successful homogeneous, catalyst-free transesterification reaction on cellulose in [mTBNH][OAC]. Moreover, [mTBNH][OAC]+DMSO mixtures can be recycled via precipitation of by-products with volatile solvents and consequent removal of solvents and used again for the reaction.

#### Ionic Liquid Recycling

The usage of ionic liquid for cellulose dissolution is beneficial regarding green chemistry concepts, but also, ILs could be regenerated and recycled to make it applicable under circular economy requirements. In the current study, the first trial to recover [mTBNH][OAC]+DMSO for further transesterification has been conducted. This is the first experimental attempt to recycle [mTBNH][OAC] after a transesterification reaction.

For the regeneration of [mTBNH][OAC]+DMSO binary solvent from the waste solution, the following procedure was established:Evaporation of water and ethanol through thin film or rotary distillation–waste pre-concentrate.Waste pre-concentrate mixed with acetone in a 1:8 ratio–hydrophilic by-products precipitate.Filtration of mixture and evaporation of acetone–waste concentrate obtained.Waste concentrate mixed with distilled water in 1:10–hydrophobic by-products precipitate.Filtration of mixture and evaporation of acetone–regenerated IL:DMSO binary solvent (recyclate) obtained.

FTIR and solubility tests were used to analyze precipitated by-product: hydrophobic precipitant is insoluble in water and common polar solvents (alcohols) and has a strong methylene group absorption peak in the FTIR spectrum; hydrophilic precipitant is insoluble in acetone and common non-polar solvents (toluene, hexane), showing strong absorption in the hydroxyl group band. 

The regenerated binary solvent has been used for the dissolution of cellulose with subsequent transesterification with vinyl laurate. Cellulose was successfully dissolved and its viscosity was identical to that of cellulose in fresh [mTBNH][OAC]:DMSO mix. Next, transesterification has been conducted under reaction conditions similar to CL-3. It was confirmed by NMR that the obtained product is CL with DS = 0.58, which is close to the value of DS = 0.6 obtained for CL-3. This trial confirms the efficiency of regenerated [mTBNH][OAC] usage for transesterification.

### 3.2. Physical Properties of Cellulose Esters

#### 3.2.1. DSC

All studied cellulose esters show similar thermal behavior in general. DSC curves for CL-10, CM, CP-3, and CS-2 in the range 1 < DS < 2 are shown in Figure 4. 

As could be seen in the plot, only CS-2 shows a strong peak at 53 °C, which corresponds to the melting of fatty acid chain crystals. All samples have glass transitions in the range 170–177 °C, which is comparable to the values reported in the literature [50].

The best DSC curves were obtained for the cellulose palmitate series (see Figure 5), where DS varied from 1 to 1.8. The specimen with DS ≈ 1 represents only fatty acid melting at 96 °C; the specimen with DS = 1.30 shows transient behavior; melting of ester chains at 108 °C coexists with a glass transition of cellulose ester chain above 160 °C. The sample with high DS = 1.80 shows a well-distinguishable glass transition temperature at 170 °C.

This kind of behavior evidences a significant change in the chain structure of cellulose esters with an increase of DS; at low DS, probably transesterification is not uniform, and the material contains a highly substituted phase and regenerated cellulose phase. The ester-rich crystalline phase has a melting point of around 100 °C. The melting peak is very broad, which could be evidence of strong variation in these crystalline phase structures.

An increase in DS leads to uniform ester chain distribution along the cellulose backbone. An amorphous phase on esterified cellulose is formed and can be observed when glass transitions near 180 °C at the DSC curve. In addition, films made of low substituted cellulose esters shrink significantly upon heating above 150 °C, which makes DSC curves noisy. Strong variation in crystalline and amorphous phase structure hinders the observation of phase transitions. Unfortunately, it was not possible to trace phase transitions regarding DS for all types of materials, only for cellulose palmitate.

#### 3.2.2. Rheology

To understand the thermoplastic properties of materials, the melt flow behavior was examined. The behavior of melt flow at low strains has a direct correlation to the macromolecular structure, molecular mass, and mass distribution of the polymer. The strain dependence of storage and loss moduli is used to evaluate thermoplastic material processability and future performance.

The dependences of storage modulus G´ and damping factor tanδ on strain are depicted in Figure 6. As can be seen in the plots, the damping factor for highly substituted material is below one in the whole measured range, while materials with DS ≤ 1.3 pass a crossover point from the rubbery region (where elastic deformations of the material dominate) at low strains to the transition region (where plastic deformations become dominant) at higher strains. The storage modulus for low-substituted CP materials drops significantly at low strains, and for the high-substituted one G´ drops above 10%. The cellulose backbone plays the main role in the viscoelastic properties of the studied materials. One could conclude that in low DS materials, flexible fatty acid branches form a separate phase with a low melting point, and this phase works as an internal plasticizer for cellulose esters. For comparison, commercially available injection-molding grade plasticized cellulose acetate by Mazzucchelli was studied. As can be seen, the shape of curves for low DS materials is similar to those for Mazzucchelli PCA, which strengthens the hypothesis of internal plasticizing by fatty acid branches.

The material with high DS appears as a homogeneous amorphous phase where fatty acid chains are evenly distributed over the cellulose backbone. The plasticizing effect of branches became less pronounced, and the material underwent only elastic deformations in all studied strain ranges.

To summarize the strain-dependent flow, the elastic deformations of the cellulose backbone, a fluctuation of physical crosslinking points, or junctions, are dominating at low strains, while the entanglement density remains constant. Fatty acid chains bonded to cellulose macromolecules are involved in the local orientation motion. A higher strain induces the inelastic motion of chains which increases the orientation of macromolecules and leads to energy dissipation due to friction. Friction significantly increases the damping factor for low-substituted materials. High-substituted materials are prone to decreasing the entanglement density only (junction points move away from each other) in the analogy with extended interconnected springs.

The viscoelastic properties of thermoplastics strongly depend on deformation (dynamic oscillations) frequency. In the frequency range between 0.1 and 100 s^−1^, studied samples of CP curves represent a rubbery plateau (see Figure 7 left), where storage prevails over loss modulus. The crossover point from the rubbery plateau to the transition region for material with the lowest DS is marked with the red arrow on the plot. At the same time conventional thermoplastic materials, like LDPE and plasticized cellulose acetate (see Figure 7 right), in this frequency range have crossover from transition region to glassy region (crossover point labeled with red arrows). As could be seen, flexible chains of LDPE and externally plasticized rigid chains of cellulose acetate at high frequencies show perfect elastic behavior and turn viscoelastic upon frequency decrease (transition zone). Cellulose palmitates in the whole frequency range show rubbery behavior due to the strong entanglement of macromolecules, which is similar to cross-linked materials. It is worth noting that CP with DS = 1.08 has crossover from rubbery to transition zone at a relatively high frequency—around 10 s^−1^; these short-range motions could be attributed to plastic deformations of fatty ester branches.

One could conclude that the melt flow of CP materials is stipulated by a rigid backbone of the cellulose chain that creates many junctions. Furthermore, material could contain solid phases of regenerated cellulose. In addition, the structural mobility could be attributed to the ester chain phase, which works as a self-lubricant for CP macromolecules in the melt. From the applied point of view, rubbery behavior makes the processing of thermoplastic material complicated. Therefore, to bring cellulose ester materials into the lower storage modulus, according to the time–temperature superposition theory, longer times for deformation, or equivalent higher temperatures, should be used. Time–temperature superposition is valid for linear homopolymers, but it highlights the approaches for materials development. A higher degree of substitution and shorter macromolecular chains should decrease entanglement density. In addition, an increase of free space within the material could cause it to soften, which could be achieved by the addition of plasticizers.

Dynamic viscosities vs. shear rate dependencies are shown in Figure 8. All samples show non-Newtonian behavior in the whole range of shear rates—no linear viscosity range could be observed. We assume that the combination of a rigid cellulose backbone with flexible fatty acid branches creates dynamic structures in the melt that are very sensitive to applied shear. The rise of the shear elastic response of the backbone, overlapped by plastic deformation of branches, leads to a continuous decrease in viscosity. Externally plasticized cellulose acetate (Mazzucchelli PCA) shows a very similar flow curve.

To summarize the rheological properties of the materials, we could conclude that two phases are present in the material formed by rigid cellulose backbone and flexible ester chains. Low-substituted material contains two distinguishable phases (probably regenerated cellulose and blocks of fatty acid chains), and high-substituted material consists of a single phase (uniformly substituted cellulose). A significant increase in relaxation times upon an increase in the degree of substitution could be caused by flexible ester chain phase elimination.

## 4. Conclusions

The present study presents an optimized FACE production method by cellulose transesterification in a novel ionic liquid, [mTBNH][OAC]. One of the important contributions of the current study is a proposed eco-friendly, catalyst-free procedure, which is a prospective alternative to commercial esterification. IL can be recycled and reused, no harmful catalyst is required, and reactants are consumed in a better stoichiometric balance. The use of a co-solvent was found to be efficient in decreasing the amount of expensive IL.

The reported approach allows the production of FACEs with selectively tuned DS from 0 up to 1.8 by controlling the following reaction conditions: the IL:DMSO ratio, temperature, and time. Moreover, the reported path allows solubility and thermoplastic properties of the end-product tuning by DS control.

It was found that DS can be increased by increasing time, temperature, and molar ratio. However, the temperature of the reaction should not exceed 80 °C and the time of reaction should be less than 5 h to avoid degradation processes. It was proved that the OH group in cellulose AGU is substituted in the order of C7-O6 > C7-O2 > C7-O3.

A second important contribution of the work is that it provides a novel, previously unreported correlation between the melt flow behavior, structure, and the degree of substitution of FACEs. At low DS, FACEs have a two-phase structure that consists of fatty ester crystalline blocks and regenerated cellulose. This type of material has a semi-solid structure and shows thermoplasticity due to the internal plasticization and self-lubricating effect of ester chains.

At high DS, CEs form a single amorphous phase that consists of uniformly esterified cellulose macromolecules. Furthermore, high-substituted materials show strong elastic response due to high junction-point density, which makes material melt flow behavior elastomer-like.

The synthesized FACEs are promising for packaging applications and could be used to substitute commercially available cellulose acetate plasticized with diethyl phthalate.

## Data Availability

The data presented in this study are available on request from the corresponding author. The data is not publicly available due to the confidentiality of the running project.

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
