# Peer review of "Preparation of Thermoplastic Cellulose Esters in [mTBNH][OAC] Ionic Liquid by Transesterification Reaction"

_polymers, 2023, doi:10.3390/polym15193979_

Round 1
Reviewer 1 Report
The paper is good in the synthesis of FACE. Tha mehod and data are good and sufficient to support the objectives.There are some thing need to discribe in detail.
1. in line 35, "is done " is not clear.
2. line 41, the word " problematic." may be problem.
3. line 66, long activation is not clear.
4. line 269, power may be ability.
5. line 275, the viscosity of cellulose solution in 1:3 IL:DMSO binary solvent is approximately 7 times lower than that of pure IL. The sentence is not clear.
6. line 330, drops 1.5 times is not clear.
1. in line 35, "is done " is not clear.
2. line 41, the word " problematic." may be problem.
3. line 66, long activation is not clear.
4. line 269, power may be ability.
Author Response
Reviewer 1
The paper is good in the synthesis of FACE. Tha mehod and data are good and sufficient to support the objectives. There are some thing need to discribe in detail.
- in line 35, "is done " is not clear.
Answer: improved
- line 41, the word " problematic." may be problem.
Answer: changed
- line 66, long activation is not clear.
Answer: activation is removed - long reaction time
- line 269, power may be ability.
Answer: changed
- line 275, the viscosity of cellulose solution in 1:3 IL:DMSO binary solvent is approximately 7 times lower than that of pure IL. The sentence is not clear.
Answer: improved
- line 330, drops 1.5 times is not clear.
Answer: improved
Comments on the Quality of English Language
- in line 35, "is done " is not clear.
Answer: improved
- line 41, the word " problematic." may be problem.
Answer: changed
- line 66, long activation is not clear.
Answer: activation is removed - long reaction time
- line 269, power may be ability.
Answer: changed

Reviewer 2 Report
Krasnou and his colleagues have submitted an article titled "Preparation of Thermoplastic Cellulose Esters in [mTBNH][OAC] Ionic Liquid by Transesterification Reaction". The functionalization of cellulose through direct and straightforward methods is considered a practical and efficient approach in both academia and industry. The final materials were analyzed and characterized appropriately.
In this project, the authors utilized ionic liquids as nonvolatile, stable, and effective media. However, the specific type of IL used is not commercially available and may be costly. Therefore, the authors should describe how it was synthesized, its price, and its thermal stability. They should also compare these parameters with previous conditions to determine its practicality from an industrial perspective.
Additionally, the authors should reference new reports on this subject, such as the two recently published articles by Takahashi et al. (DOI: https://doi.org/10.1039/D2RE00537A and DOI: 10.1039/D2GC04730F). It is crucial to note that in the transesterification process, the carbonyl carbon of the starting ester reacts to give a tetrahedral intermediate, which either reverts to the starting material or proceeds to the transesterified product.
Finally, it is highly recommended that the authors analyze the XRD characterization of cellulose and the final material.
Krasnou and his colleagues have submitted an article titled "Preparation of Thermoplastic Cellulose Esters in [mTBNH][OAC] Ionic Liquid by Transesterification Reaction". The functionalization of cellulose through direct and straightforward methods is considered a practical and efficient approach in both academia and industry. The final materials were analyzed and characterized appropriately.
In this project, the authors utilized ionic liquids as nonvolatile, stable, and effective media. However, the specific type of IL used is not commercially available and may be costly. Therefore, the authors should describe how it was synthesized, its price, and its thermal stability. They should also compare these parameters with previous conditions to determine its practicality from an industrial perspective.
Additionally, the authors should reference new reports on this subject, such as the two recently published articles by Takahashi et al. (DOI: https://doi.org/10.1039/D2RE00537A and DOI: 10.1039/D2GC04730F). It is crucial to note that in the transesterification process, the carbonyl carbon of the starting ester reacts to give a tetrahedral intermediate, which either reverts to the starting material or proceeds to the transesterified product.
Finally, it is highly recommended that the authors analyze the XRD characterization of cellulose and the final material.
Author Response
Reviewer 2
Krasnou and his colleagues have submitted an article titled "Preparation of Thermoplastic Cellulose Esters in [mTBNH][OAC] Ionic Liquid by Transesterification Reaction". The functionalization of cellulose through direct and straightforward methods is considered a practical and efficient approach in both academia and industry. The final materials were analysed and characterized appropriately.
In this project, the authors utilized ionic liquids as nonvolatile, stable, and effective media. However, the specific type of IL used is not commercially available and may be costly. Therefore, the authors should describe how it was synthesized, its price, and its thermal stability. They should also compare these parameters with previous conditions to determine its practicality from an industrial perspective.
Answer: Synthesis and stability of used IL are discussed in Introduction. Synthesis is described in (Martins, 2022), thermal and hydrolysis stability is described in Ostonen, 2016).
We cannot provide information about production price so far as it is a trade secret of Liuotin company. Considering the price question, we offer to use DMSO as a co-solvent, the total price for the solvents will be twice lower. In any case to be used in industrial upscaling novel IL should be comprehensively studied.
The scope of this work is in optimization of transesterification reaction in ([mTBDH][OAc]) but not study of IL properties. IL properties are studied and compared in papers (King, 2011; Elsayed, 2020). These questions are briefly discussed in Introduction.
Additionally, the authors should reference new reports on this subject, such as the two recently published articles by Takahashi et al. (DOI: https://doi.org/10.1039/D2RE00537A and DOI: 10.1039/D2GC04730F). It is crucial to note that in the transesterification process, the carbonyl carbon of the starting ester reacts to give a tetrahedral intermediate, which either reverts to the starting material or proceeds to the transesterified product.
Answer: Thank you for suggestion, we will include these articles into Introduction part.
Finally, it is highly recommended that the authors analyze the XRD characterization of cellulose and the final material.
Answer: Thank you for recommendation. XRD characterization will be included in next article, which deals with microstructure investigation of prepared materials. Current article is applied study which stresses the properties and conditions for synthesis of FACE, and thermoplastic properties important for material processing and application.

Reviewer 3 Report
Review Report:
Manuscript Number: polymers-2622951
Title: Preparation of Thermoplastic Cellulose Esters in [mTBNH][OAC] Ionic Liquid by Transesterification Reaction
Recommendation: Minor revision
The paper " Preparation of Thermoplastic Cellulose Esters in [mTBNH][OAC] Ionic Liquid by Transesterification Reaction " named and polymers-2622951 coded manuscript. In this work, the authors provides a clear and concise overview of the study's objectives, methodology, and key findings. It effectively communicates the significance of the research, highlighting the potential environmental benefits of using ionic liquids for cellulose ester synthesis. Overall, the study is well-conducted and provides valuable insights into the properties of the compound. However, there are several areas that need improvement and clarification before the paper can be considered for publication. The major issues are outlined below:
1. While the abstract mentions the use of distillable ionic liquid [mTBNH][OAc], it would be helpful to briefly explain what this ionic liquid is and why it is suitable for the transesterification reaction. Providing a brief context for non-specialist readers can enhance comprehension.
2. Could you provide more details about the transesterification reaction conditions, such as the types of vinyl esters used and the specific reaction temperatures and times employed? Additionally, it would be helpful to explain why DMSO was chosen as the co-solvent and its role in the reaction.
3. The abstract mentions the synthesis of cellulose esters with a degree of substitution up to 1.8 and the optimization of various reaction parameters. It would be beneficial to include specific quantitative results or trends observed in the optimization process to give readers a better sense of the study's outcomes.
4. The authors mentions FTIR, 1H, and 13C NMR for confirming the product structure. Could you briefly describe any noteworthy spectroscopic results or structural changes observed in the cellulose esters during the study?
5. Are there any specific applications or industries where these long-chain cellulose esters might find immediate use, and do you plan to explore their practical applications in future research?
6. Recycling of Ionic Liquid: Since the ionic liquid [mTBNH][OAc] is mentioned as recyclable, could you briefly explain the recycling process and any challenges associated with it?
7. The manuscript contains numerous grammatical errors and awkward sentence structures. Athorough proofreading is necessary to improve the overall clarity and readability of the paper.
8. The conclusion should summarize the main findings of the study and their implications. It should also highlight the novelty and significance of the work.
Overall, the abstract effectively conveys the essence of the research but could benefit from additional details and clarification of certain points to enhance its comprehensibility and appeal to a broader audience.

Acceptable
Author Response
Reviewer 3
Manuscript Number: polymers-2622951
Title: Preparation of Thermoplastic Cellulose Esters in [mTBNH][OAC] Ionic Liquid by Transesterification Reaction
Recommendation: Minor revision
The paper " Preparation of Thermoplastic Cellulose Esters in [mTBNH][OAC] Ionic Liquid by Transesterification Reaction " named and polymers-2622951 coded manuscript. In this work, the authors provides a clear and concise overview of the study's objectives, methodology, and key findings. It effectively communicates the significance of the research, highlighting the potential environmental benefits of using ionic liquids for cellulose ester synthesis. Overall, the study is well-conducted and provides valuable insights into the properties of the compound. However, there are several areas that need improvement and clarification before the paper can be considered for publication. The major issues are outlined below:
- While the abstract mentions the use of distillable ionic liquid [mTBNH][OAc], it would be helpful to briefly explain what this ionic liquid is and why it is suitable for the transesterification reaction. Providing a brief context for non-specialist readers can enhance comprehension.
Answer: The appropriate sentences about IL were included into Abstract.
- Could you provide more details about the transesterification reaction conditions, such as the types of vinyl esters used and the specific reaction temperatures and times employed? Additionally, it would be helpful to explain why DMSO was chosen as the co-solvent and its role in the reaction.
Answer: Specific details about the transesterification reaction conditions (types of vinyl esters, reaction temperatures and times) were included into abstract according to the comment.
- The abstract mentions the synthesis of cellulose esters with a degree of substitution up to 1.8 and the optimization of various reaction parameters. It would be beneficial to include specific quantitative results or trends observed in the optimization process to give readers a better sense of the study's outcomes.
Answer: The specific quantitative trends in DS related to reaction conditions have been added to Abstract.
- The authors mentions FTIR, 1H, and 13C NMR for confirming the product structure. Could you briefly describe any noteworthy spectroscopic results or structural changes observed in the cellulose esters during the study?
Answer: The major findings of FTIR and NMR results along with the possible structure (substitution) of cellulose esters have been included to Abstract
- Are there any specific applications or industries where these long-chain cellulose esters might find immediate use, and do you plan to explore their practical applications in future research?
Answer: currently we work over scaling up the transesterification procedure what will be published later. FACE are promising materials for specialized packaging like food and drugs, due to its good barrier properties. n
- Recycling of Ionic Liquid: Since the ionic liquid [mTBNH][OAc] is mentioned as recyclable, could you briefly explain the recycling process and any challenges associated with it?
Answer: A paragraph about recycling is added to manuscript
- The manuscript contains numerous grammatical errors and awkward sentence structures. Athorough proofreading is necessary to improve the overall clarity and readability of the paper.
Answer: language is improved with a help of proofreading
- The conclusion should summarize the main findings of the study and their implications. It should also highlight the novelty and significance of the work.
Answer: Conclusions part is improved accordingly
Overall, the abstract effectively conveys the essence of the research but could benefit from additional details and clarification of certain points to enhance its comprehensibility and appeal to a broader audience.
Answer: Abstract is improved according to recommendations

Round 2
Reviewer 2 Report
The article could be accepted in this present.